# Application of β-Glucosidase in a Biphasic System for the Efficient Conversion of Polydatin to Resveratrol

**DOI:** 10.3390/molecules27051514

**Published:** 2022-02-23

**Authors:** Jie Zhou, Meng Liang, Yu Lin, Hao Pang, Yutuo Wei, Ribo Huang, Liqin Du

**Affiliations:** 1State Key Laboratory for Conservation and Utilization of Subtropical Agro-Bioresources, Guangxi Research Center for Microbial and Enzymatic Technology, College of Life Science and Technology, Guangxi University, Daxue Road No. 100, Nanning 530005, China; zhoujie@gxu.edu.cn (J.Z.); lm9267@163.com (M.L.); wslinyu2022@163.com (Y.L.); weiyutuo@gxu.edu.cn (Y.W.); guruace@163.com (R.H.); 2Guangxi Key Laboratory of Bio-Refinery, National Engineering Research Center for Non-Food Biorefinery, State Key Laboratory of Non-Food Biomass and Enzyme Technology, Guangxi Academy of Sciences, Daling Road No. 98, Nanning 530007, China

**Keywords:** resveratrol, polydatin, β-D-Glucosidase, biphasic enzymatic transformation

## Abstract

Resveratrol, an ingredient of traditional Chinese medicine, has beneficial effects on human health and huge potential for application in modern medicine. Polydatin is extracted from plants and then deglycosylated into resveratrol; enzymatic methods are preferred for this reaction. In this study, a β-D-glucosidase from *Sphingomonas* showed high efficiency in transforming polydatin into resveratrol and was tolerant toward organic solvents. Applying this enzyme in a biphasic transformation system resulted in 95.3% conversion of 20% concentration crude polydatin to resveratrol in 4 h. We thus report a new method for high-efficiency, clean production of resveratrol.

## 1. Introduction

Stilbenes are secondary metabolites of plants. Because of their potential effects in ameliorating human health [1,2,3], increasing attention has been paid to their synthesis and molecular modes of action. Among the many stilbenes, resveratrol (*trans*-3,5,4 trihydroxystilbene) is the most prominent in medical applications. Resveratrol is believed to have anti-inflammatory [4], anti-aging [5] and cardioprotective effects [6], and it has been applied as nutritional supplement in health products and cosmetics products [7]. Resveratrol is found in many plants such as vegetables [8] and fruits [9,10,11,12]. The root of *Polygonum cuspidatum* was found to be the richest source of resveratrol [13]. Compared with resveratrol synthesized by chemical synthesis (de novo synthesis), resveratrol produced by the *P. cuspidatum* extraction process is more economical and has been widely used.

In plants, resveratrol is present as a glycoside, polydatin, in which a glucoside group bound to the C-3 position substitutes for a hydroxyl group (Figure 1). However, polydatin has lower bioavailability than resveratrol, and it is shown that human intestinal cell absorb polydatin with greater difficulty and more slowly than resveratrol [14,15]. Thus, deglycosylation of polydatin is important to obtain the active resveratrol. In recent years, many new methods have been developed for producing resveratrol from polydatin [16,17,18,19,20,21]. Bioconversion is a promising approach because of its high conversion efficiency and environmental friendliness compared with conventional chemical synthesis and hydrolysis methods. It is reported that diverse fungi show great potential in converting polydatin to resveratrol, including *Aspergillus niger* [22], *Rhizopus* sp. T-34 [23] and other fungi [24]. However, 5–8 days are generally needed to attain high-level conversion.

Polydatin can be converted to resveratrol by β-D-glucosidases. Direct enzymatic conversion of polydatin to resveratrol should be more efficient and less time consuming than microbial whole cell conversion because the reaction conditions can be set to match the optimal conditions for catalysis by the enzyme. However, for glycoside aromatic compounds containing aglycones, the large aglycone spatially blocks the contact of the enzyme to the glycosidic bond. It is rare that an enzyme has the ability to hydrolyze this linkage between aryl aglycones and glycosyl group [25,26]. Chen et al. [27] reported that a piceid-β-D-glucosidase cloned from *Aspergillus* showed preferable characteristics and high overall conversion efficiency in converting polydatin to resveratrol. Bacteria are considered to be more suitable for the biotransformation of polydatin than mold, but the enzyme suitable for the transformation of polydatin screened from bacteria is lacking [17]. 

The poor aqueous solubility and good organic solvent solubility of polydatin and resveratrol result in another complication in both enzymatic and whole-cell conversions. Generally, conversion reactions are carried out in aqueous solutions with little or no organic solvent [19,21]. Subsequently, the substrate and the product are mixed and precipitated, and the product must be recovered using organic solvent extraction [19]. Unavoidably, the final product will be contaminated by substrate. 

To overcome the conversion problem of resveratrol, a biphase transformation system was set up using a β-D-glucosidase, named SpBGL1, from *Sphingomonas* sp. ATCC31555. This enzyme showed preferable characteristics in converting polydatin to resveratrol and was highly tolerant toward organic solvents. This biphase biotransformation system resulted in good conversion of polydatin to resveratrol and high product purity.

## 2. Results 

### 2.1. Enzymatic Properties of β-D-Glucosidase SpBGL1

A putative glycoside hydrolase family (GH) 3 β-D-glucosidase-encoding gene (WP_019370396.1) was identified in the genome of *Sphingomonas* sp. ATCC 31555 by sequence analysis. The product of this gene was named SpBGL1. The deduced protein has 802 amino acids (Figure 2); the predicted molecular mass and pI were calculated as 86.911 kDa and 6.76, respectively by using Geneious 11.0.5.

SpBGL1 hydrolyzed *p*NPG (4-Nitrophenyl-β-D-glucopyranoside), proving that this enzyme is a β-D-glucosidase. When using *p*NPG as the substrate, the enzyme showed its highest hydrolytic activity at pH 5.5 at 45 °C; the K_m_ and V_max_ values were 13.09 ± 3.43 mM and 5.817 ± 0.59 μmol/min, respectively.

SpBGL1 showed high tolerance to glucose. The tolerance was glucose-concentration dependent. SpBGL1 maintained 55% relative activity in the presence of 1 M glucose, while in the presence of 3 M glucose, 28% relative activity was maintained. The K_i_ value calculated using the SNLR (simultaneous nonlinear regression fit) method was 60.9 ± 0.45 (Figure 3). 

Table 1 was determined (Table 1). In the optimum reaction conditions, SpBGL1 showed good efficacy in hydrolyzing epimedin A, icariin, daidzin and polydatin. 

The tolerance of SpBGL1 to diverse organic compounds was investigated. SpBGL1 showed good tolerance to methanol and ethanol; relative enzyme activity of 60.1% and 90.9% was maintained in the presence of 15% (*v*/*v*) methanol and ethanol, respectively, and 69.1% relative activity was maintained in 20% (*v*/*v*) ethanol (Figure 4). Furthermore, Tween 80 (5%, *v*/*v*) showed negligible inhibition of the activity of SpBGL1, and Triton X-100 (1%, *v*/*v*), β-mercaptoethanol (1%, *v*/*v*) and imidazole (10 mM) resulted in slight enhancement of the activity (Figure 5).

### 2.2. Optimal Reaction Conditions for Hydrolyzing Polydatin

Assays were carried out to determine the optimal reaction temperature and pH for conversion of polydatin into resveratrol by SpBGL1. The relative enzymatic activity of SpBGL1 was measured at 40 °C for 20 min at pH values ranging from 4.5 to 7 (interval 0.5). The maximum conversion ratio, 67.4%, was observed at pH 6.0 (Figure 6A). 

To determine the effect of temperature at the optimal pH, enzymatic conversions were carried out at 35, 40, 45, 50, 55 and 60 °C and pH 6.0 for 30 min. The maximum conversion ratio was observed at 40 °C. At 60 °C, only 10.07% of the polydatin was transformed into resveratrol (Figure 6B).

### 2.3. Comparison of Uniaphase and Biphase Enzymatic Conversion for Polydatin Hydrolysis

Two different enzymatic transformation systems were set up and compared for their effectiveness in converting polydatin to resveratrol (Figure 7). The reactions in both the uniphase and the biphase systems were carried out on polydatin extract (20% purity), final concentration 20 g/L, at 40 °C and pH 6.0. As shown in Figure 7, in the biphase system, the color of the organic phase darkened as the reaction took place, and the amount of crude polydatin decreased dramatically (Figure 8A). The amount of crude polydatin did not show an observable decrease (by eye) in the uniphase conversion system (Figure 8B). The reaction solutions in both systems were analyzed by high-performance liquid chromatography (HPLC) at different time points to determine the overall conversion ratio (Figure 9). At all reaction time points, the conversion ratio in the biphase system were significantly higher than those in the uniphase system. In the biphase system, the overall conversion ratio reached 95.25% in 4 h, whereas the conversion ratio reached only 85.74% at 6 h in the uniphase system (Figure 10).

### 2.4. Optimization of Biphase Enzymatic Conversion System

Based on the results described above, the biphase conversion system was chosen for further study. To investigate the optimal final concentration of polydatin extract (20% purity), reactions containing 20, 30, 40 and 50 g/L of crude polydatin were carried out at 40 °C and pH 6.0. As shown in Figure 11, a high conversion ratio was observed in the reactions containing 20 and 30 g/L substrate. To maximize resveratrol production, 30 g/L was chosen as the optimum substrate concentration. 

The ability of SpBGL1 to convert crude polydatin (20% purity) to resveratrol in the optimum conditions was determined. As shown in Figure 12, after 1 h, 71.25% of the substrate was hydrolyzed, and after 4 h the substrate was almost completely hydrolyzed (95.3%). The deduced resulting concentration of resveratrol was 3.34 g/L.

## 3. Discussion

Bioconversion of polydatin to resveratrol is advantageous compared with other conversion methods, such as acid hydrolysis, in terms of energy consumption, complexity and resulting pollution. Microbial conversion was once the most widely used method for bioconversion of polydatin [28]. However, the main drawback of this method is that the culture conditions of the microorganisms, such as temperature and pH, are different from those for the optimal function of the enzyme. Second, the fermentation generally requires a relatively long time to achieve a high conversion ratio. Hence, enzymatic conversion of polydatin to resveratrol could be more efficient. Unfortunately, however, few enzymes suitable for efficient industrial conversion of polydatin to resveratrol have been reported. Generally, β-D-glucosidases from bacteria are less efficient than those originating from fungi, such as the piceid-β-D-glucosidase purified from *A. oryzae* sp. 100, which can convert 90% of polydatin to resveratrol in 4 h [27]. In the work of Hu et al., [17] several glucosidases from *Bacillus safensis* were studied in conversion of polydatin to resveratrol; only two showed outstanding activity toward polydatin, and the best converted 93.1% of the resveratrol in 8 h when the substrate concentration was between 0.1% and 0.3%. 

In this study, we cloned and characterized a GH3 family β-D-glucosidase, SpBGL1, from *Sphingomonas* sp. ATCC31555. In previous work, we found that the enzymes derived from the *Novosphingobium* sp strain had special action on polyphenol glycoside substrates [29]. We hypothesized that there are similar enzymes in these related strains. These enzymes differ from other glycoside hydrolases in that they have the ability to act on glycoside compounds with aromatic aglycone. Indeed, SpBGL1 shows the ability to achieve that. This enzyme showed preferable characteristics for efficient conversion of polydatin into resveratrol. By applying SpBGL1 in a biphasic conversion system, the conversion ratio of 20% concentration crude polydatin to resveratrol reached 95.3% in 4 h. 

Polydatin and resveratrol are both poorly soluble in water; thus, microbial and enzymatic conversions generally require a secondary extraction step to recover the resveratrol. In this study, a minimized biphasic enzymatic transformation system was set up to overcome this drawback. Polydatin is poorly water-soluble, unable to dissolve at 20%, but can be enzymatically hydrolyzed in this system. Resveratrol produced during the course of the conversion continuously dissolved in the organic phase, which allowed recovery of the product by simply removing the organic phase and vacuum evaporation. This procedure is less laborious and more effective compared to the extraction procedures that are usually required in uniphase conversion systems. As described by Huang et al., [30] the resveratrol produced in the uniphase system requires organic solvent recovery followed by sonication and multiple suction filtration steps to remove impurities before the final product precipitation and vacuum evaporation. 

Raw extracts of polydatin were used as the substrate to test the efficiency of the system. The conversion ratio of polydatin by SpBGL1, 95.3% in 4 h, was comparable to that obtained in the conventional conversion system while obtaining resveratrol with high purity without the need for extra organic solvent extractions. The system is capable of decreasing the pretreatment of the raw material, lowering the requirement for crude substrate purity while maintaining the overall conversion efficiency. Since, generally, a large reaction volume is preferable for enzymatic conversions, SpBGL1 shows huge potential for industrial resveratrol production. Other than the application in polydatin conversion as reported in this study, the minimized biphasic enzymatic transformation system set up here could be applied to small-scale enzymatic reactions of other raw materials that have poor water solubility, which is often the case for raw materials in Chinese medicine.

## 4. Materials and Methods

### 4.1. Bacterial Strains and Growth Conditions

*Sphingomonas* sp. ATCC 31555 was purchased from the American Type Culture Collection (ATCC). Strain ATCC31555 was grown in YPG medium (5 g of tryptone, 3 g of yeast extract and 20 g of glucose per liter of H_2_O) at 37 °C in a shaker at 200 rpm. All *Escherichia coli* strains were cultured in LB medium (10 g of tryptone, 5 g of yeast extract and 10 g of NaCl per liter of H_2_O) at 37 °C.

### 4.2. Cloning of SpBGL1

The open reading frame encoding SpBGL1 (accession number: WP_019370396.1) was amplified using primers 96Fw (agcttcatgatgCAGACCCAGGCCGCCCGGGC) and 96Rev (agctaagcttAGTGGTGATGATGGTGATGTCGAACGGTGAGCGTGGC). The forward primer was designed to avoid amplifying the signal peptide and a start codon was added immediately upstream of the 29th amino acid. Motif analysis was performed using SMART (http://smart.embl-heidelberg.de/, accessed on 10 January 2022). *Nco*I and *Hin*dIII restriction sites were included at the 5′- end of the forward and reverse primers, respectively. The *Nco*I and *Hin*dIII-digested PCR product was ligated with pQE30 digested with the same enzymes, resulting in pQE-SpBGL1.The integrity of the cloned gene was verified by DNA sequencing.

### 4.3. Heterologous Expression of SpBGL1

pQE-SpBGL1 was transformed into *E. coli* strain JM109 by electroporation. The resulting transformants were verified using primers 96Fw and 96Rev. To express the β-glucosidase, a recombinant clone was pre-cultured overnight in 10 mL of LB medium, then 4 mL of the pre-culture were inoculated into 200 mL of LB medium supplemented with 100 mg/mL ampicillin and 0.5 mM isopropyl β-D-1-thiogalactopyranoside. The culture was incubated at 20 °C for 24 h in a shaker at 200 rpm. 

The cells in 1 L of culture were collected by centrifugation at 8000 rpm for 10 min, then resuspended in 6 mL of H_2_O and lysed by sonication (JY92-2D, Xinzhi Ltd., Ningbo, China). The cell-free lysate was loaded onto a Ni-NTA column (Qiagen, Venlo, The Netherlands). The column was washed once with 4 mL of wash buffer (NaH_2_PO_4_, 50 mM; NaCl, 300 mM; imidazole, 20 mM; pH 8.0), then the β-glucosidase was eluted with 2 mL of elution buffer (NaH_2_PO_4_, 50 mM; NaCl, 300 mM; imidazole, 250 mM; pH 8.0). The purified protein was confirmed by SDS-PAGE.

### 4.4. Enzyme Activity Assays

The activity of the purified β-glucosidase was assayed using *p*NPG as the substrate. Briefly, 10 μL of the enzyme were mixed with 20 μL of *p*NPG in 170 μL of McIlvaine buffer. The mixtures were incubated at the desired temperature for 20 min, then the reactions were terminated using 1 M Na_2_CO_3_. The absorbance of the mixture was measured at 405 nm. Each reaction was performed independently three times.

Product inhibition assays were carried out as described above except glucose (0.5–3 M, 0.5 M interval) was added to the reaction; these reactions were carried out at the optimum temperature and pH.

Organic solvent tolerance was tested in reactions containing ethanol or methanol (5–30%, *v*/*v*). Chemical compound tolerance was tested by adding Triton X-100 (1%), β-mercaptoethanol (1%), Tween 80 (5%) or imidazole (10 mM) to the reaction. All reactions were carried out at the optimum temperature and pH as described above, and the relative activities were calculated using the activity of SpBGL1 in additive-free reaction buffer as a reference.

### 4.5. Enzymatic Characterization of SpBGL1

The optimum pH of SpBGL1 with *p*NPG as the substrate was determined at 37 °C. SpBGL1 (10 μL) was mixed with 20 μL of substrate (2 mM) in the following buffers: 100 mM glycine-HCl for pH 2.0–3.0, McIlvaine buffer for pH 3.0–8.0, 100 mM Tris-HCl for pH 8.0–9.0 and 100 mM glycine-NaOH for pH 9.0–12.0. The volume of the mixtures was adjusted to 200 μL, they were incubated at 37 °C for 20 min and the absorbance was measured as described in Section 4.4.

The optimum temperature for the enzyme activity was measured at the optimum pH by measuring the enzymatic activity at 20, 25, 30, 35, 40, 45, 50, 55 and 60 °C. 

K_m_ and V_max_ values of SpBGL1 were determined at the optimum reaction temperature in McIlvaine buffer (at the optimum pH) containing 0.1–20 mM *p*NPG. The experiment was triplicated, and the values were calculated using the Lineweaver–Burk method and GraFit software (version GraphPad Prism 5, Erithacus Software, Horley, UK). The K_i_ value was calculated using the SNLR (simultaneous nonlinear regression fit) method [31].

### 4.6. Optimal Enzymatic Conditions for Catalysis of Polydatin by SpBGL1

The optimal pH for catalysis of polydatin by SpBGL1 was determined as described in Section 4.5, except that 20 μL of polydatin (100 μg/mL) were used as the substrate and the pH value was adjusted to 4.5–7.0. The optimal temperature for the reaction was determined as described in Section 4.5, except the temperature was set to 35 to 60 °C (at 5 °C intervals). 

HPLC was performed using a Waters ALLIANCE E2695 HPLC system (Milford, MA, USA) equipped with a C18 column. The isocratic elution of polydatin and resveratrol was carried out using acetonitrile (35%) at a flow rate of 0.8 mL/min. Polydatin and its hydrolysis products were detected at 303 nm. Calibration curves were established to determine the concentrations of polydatin and resveratrol. Briefly, standard solutions of polydatin (1, 2, 3, 4, 5 and 10 g/L) and resveratrol (1, 2, 3, 4, 5 and 10 g/L) were subjected to HPLC analysis and the corresponding peak area was plotted against the concentration of the standard solution using Microsoft Excel.

### 4.7. Enzymatic Hydrolysis Assay in The Uniphase and Biphasic Conversion Systems

Uniphase and biphasic transformation systems were set up to compare their effectiveness in converting polydatin into resveratrol. The biphasic enzymatic hydrolysis system was established as described by Shen et al. [32] with modifications adapted to our sample and purpose: first, the system contained 3 mL of aqueous phase (McIlvaine buffer, pH 6.0) and organic phase (ethyl acetate) each; second, SpBGL1 was added to the aqueous phase and the desired amount of crude polydatin was added to the aqueous phase. At each sampling time point, organic phase solution was carefully removed for vacuum concentration. The resulting powder was dissolved in 80% methanol and subjected to HPLC analysis for the quantification of resveratrol. The uniphase system was composed of 3 mL of McIlvaine buffer (pH 6.0). In both systems, crude polydatin (20% purity) was added into the reaction without preheating to increase the solubility (in form of precipitation), the final concentration of SpBGL1 was 180 μg/mL and the reactions were conducted at 40 °C. Recovery of resveratrol from the uniphase conversion system was carried out according to Huang et al., except that methanol was used to recover the resveratrol before vacuum concentration [30]. 

## 5. Conclusions

To efficiently convert polydatin to resveratrol, GH3 family β-D-glucosidase SpBGL1 was selected and characterized. This enzyme can effectively convert polydatin to resveratrol and has good tolerance toward organic solvents. By use of this enzyme in a biphasic system, 95.3% conversion of polydatin to resveratrol was achieved and high-purity resveratrol was obtained without additional organic solvent extraction.

## Figures and Tables

**Figure 1 molecules-27-01514-f001:**
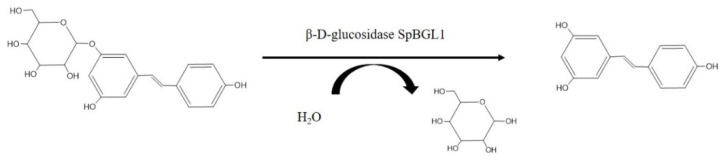
The conversion of polydatin into resveratrol by SpBGL1, a β-D-glucosidase from *Sphingomonas* sp. ATCC31555. One molecule of polydatin is hydrolyzed into one molecule of glucose and one molecule of resveratrol.

**Figure 2 molecules-27-01514-f002:**
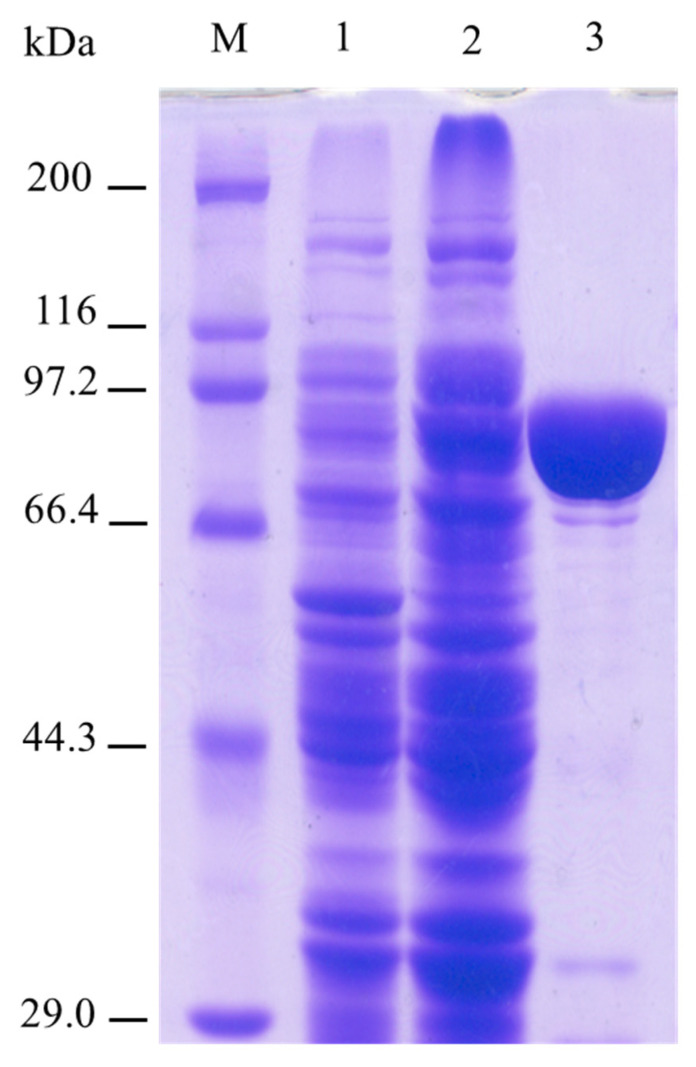
SDS polyacrylamide gel electrophoresis analysis of purified recombinant SpBGL1. M, molecular weight marker (Takara, Protein Molecuar Weight Marker (Broad), D532A); 1—crude protein extraction of JM109 harboring pQE30; 2—crude protein extraction of JM109 harboring pQE-SpBGL1; 3—purified SpBGL1.

**Figure 3 molecules-27-01514-f003:**
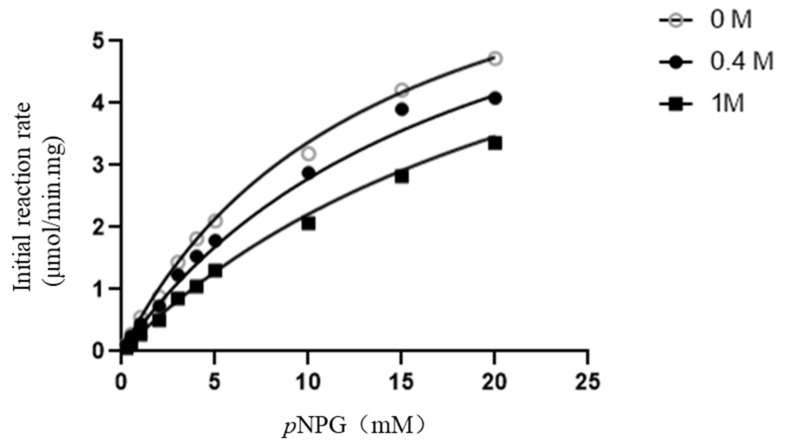
The determination of glucose inhibition constant using SNLR method.

**Figure 4 molecules-27-01514-f004:**
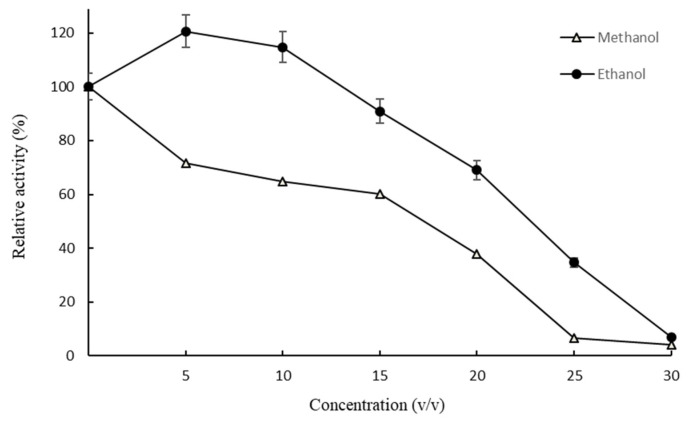
The effects of methanol and ethanol on the enzymatic activity of SpBGL1 with *p*NPG as substrate. The relative activity of the enzyme is plotted against the concentration of methanol or ethanol in the reaction solution. Each value in the figure represents the mean ± standard deviation (*n* = 3).

**Figure 5 molecules-27-01514-f005:**
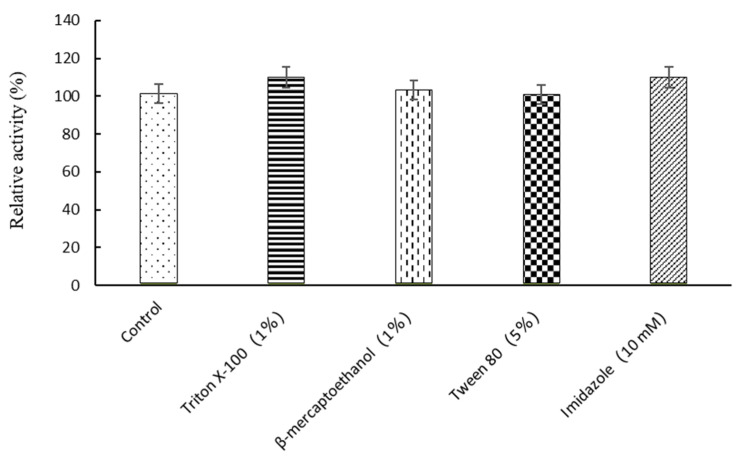
The relative enzymatic activity of SpBGL1 with *p*NPG as substrate in the presence of Triton X-100, β-mercaptoethanol, Tween 80, or imidazole. The experiments were triplicated, and the error bars represent the standard deviation.

**Figure 6 molecules-27-01514-f006:**
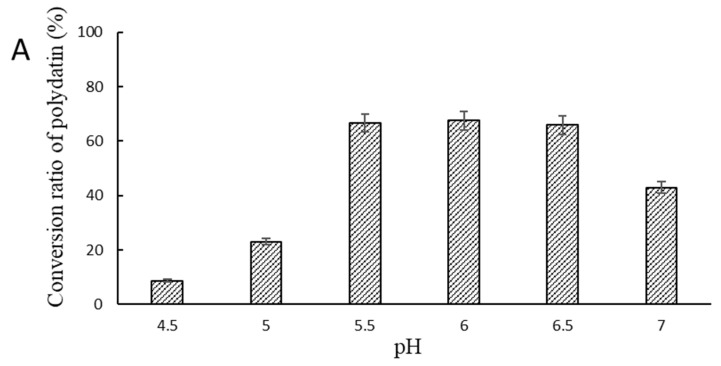
Effect of (**A**) pH and (**B**) temperature on the enzymatic activity of SpBGL1 in converting polydatin to resveratrol. The effect of pH was determined by incubating the enzyme with the substrate at 37 °C at different pH values for 20 min; the effect of temperature was determined at pH 6.0 for 20 min.

**Figure 7 molecules-27-01514-f007:**
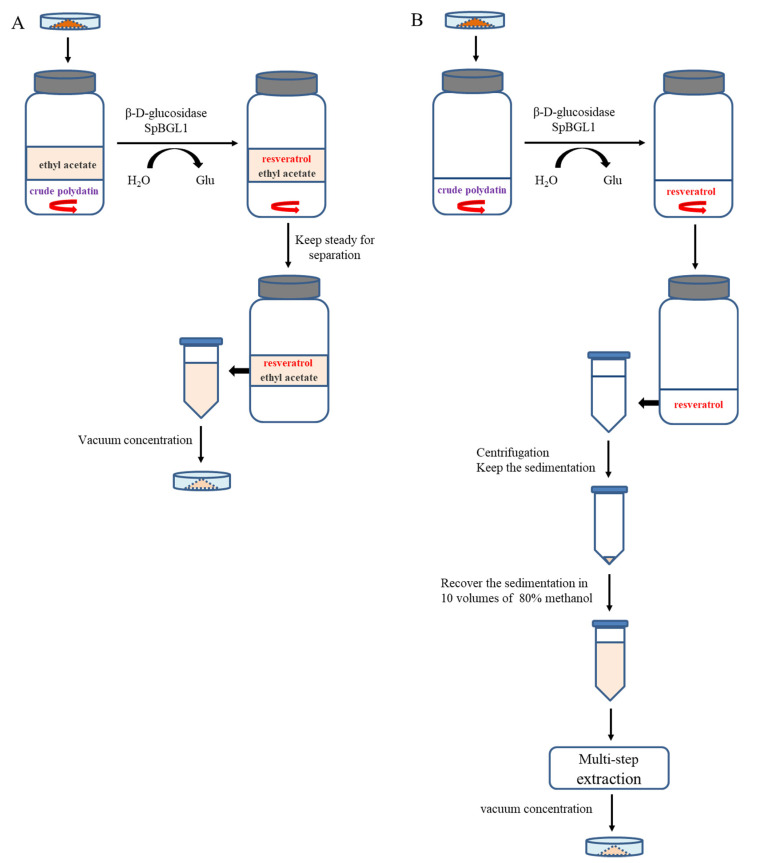
Schematic presentation of the biphase (**A**) and uniphase (**B**) conversion systems set up in this study and the major procedure of using these systems to convert polydatin into resveratrol. In the biphase system, the conversion took place in the aqueous phase, and the resulting resveratrol continuously dissolved in the organic phase. Resveratrol can be extracted directly from the organic phase by vacuum concentration. In the uniphase conversion system, because of the low solubility of resveratrol in aqueous solution, the product must first be separated from the aqueous solution by centrifugation, then recovered in 80% ethanol. This solution is then passed through an overnight multistep extraction procedure before finally being vacuum concentrated.

**Figure 8 molecules-27-01514-f008:**
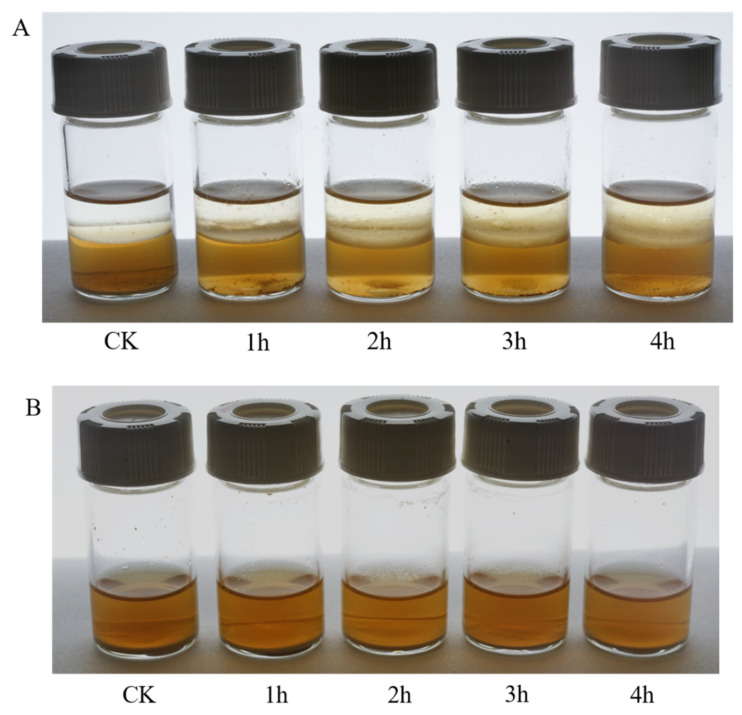
Conversion of polydatin into resveratrol by SpBGL1 in the biphase (**A**) and uniphase (**B**) conversion systems. Pictures were taken every hour after stopping agitation.

**Figure 9 molecules-27-01514-f009:**
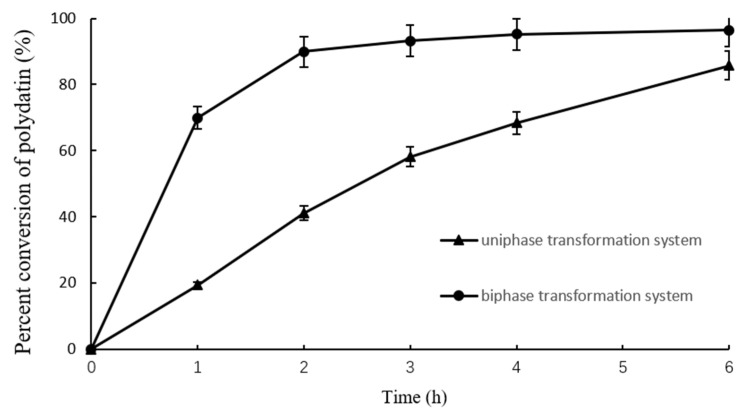
Effect of reaction time on conversion of polydatin by SpBGL1 in the uniphase and biphase conversion systems.

**Figure 10 molecules-27-01514-f010:**
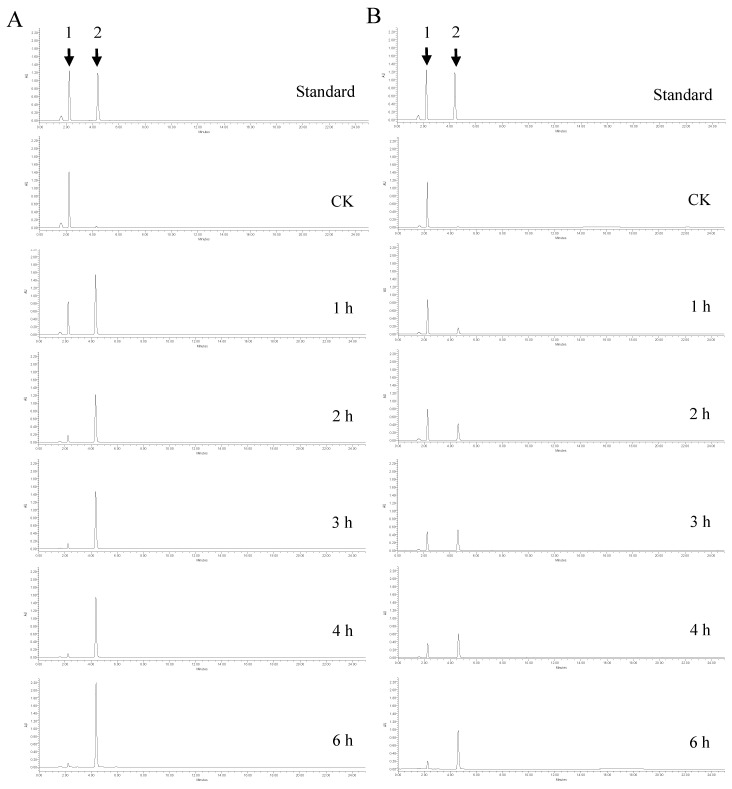
High-performance liquid chromatograms of the reaction solution from different time points in the biphase conversion system (**A**) and the uniphase conversion system (**B**). Standard, (1) polydatin; (2) resveratrol. CK, reactions without enzyme; 1–6 h, different time points after the enzyme action.

**Figure 11 molecules-27-01514-f011:**
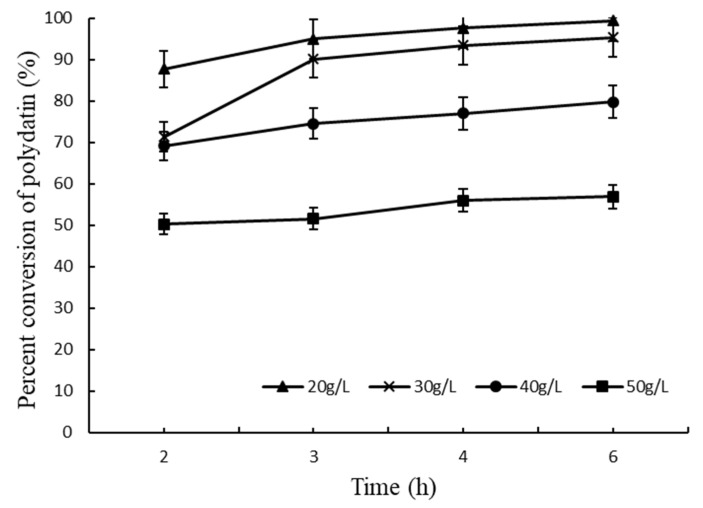
Effect of polydatin concentration on the conversion ratio. Crude polydatin (20% purity) was added to the reaction to final concentrations of 20, 30, 40 and 50 g/L.

**Figure 12 molecules-27-01514-f012:**
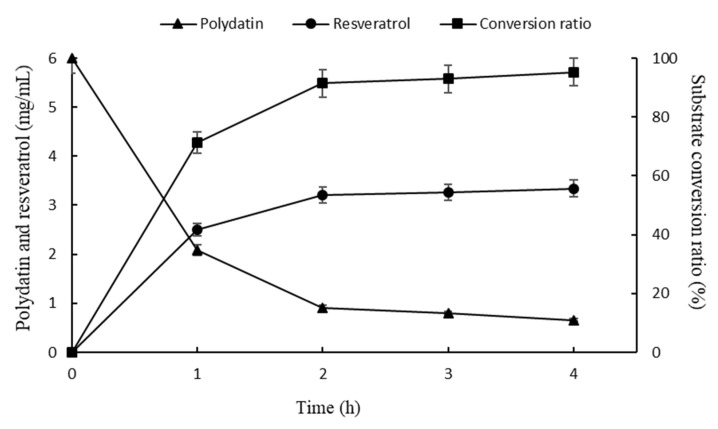
Time course of enzymatic transformation of crude polydatin and the deduced production of resveratrol. Each value in the figure represents the mean ± standard deviation (*n* = 3).

**Table 1 molecules-27-01514-t001:** Substrate profile of SpBGL1.

Substrate	Activity
Polydatin	+
Daidzin	+
Glycitin	+
Genistin	+
Icariin	+
Epimedin A	+
Epimedin B	+
Epimedin C	+
Rebaudioside A	−
Stevioside	−

Reaction by SpBGL1 was performed for 30 min.

## Data Availability

All the data supporting the findings of this study are included in this article.

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
