# Peer review of "Application of β-Glucosidase in a Biphasic System for the Efficient Conversion of Polydatin to Resveratrol"

_molecules, 2022, doi:10.3390/molecules27051514_

Round 1
Reviewer 1 Report
- The manuscript " Application of β-glucosidase in a biphasic system for the efficient conversion of polydatin to resveratrol" by Zhou and colleagues reports the cloning, the biochemical characterization, and the application of β-glucosidase in the biphasic system for converting polydatin to resveratrol.
- The work as such seems interesting and technically sound, however, there are a few issues that must be addressed before its publication.
- First, even after reading the methodology section, the setup for uniphasic and biphasic systems is not clear to me, in these systems polydatin conversion took place in the aqueous phase, though the solubility of polydatin is better than resveratrol, it has still low solubility in aqueous buffers. In this condition, I think achieving 20g/L or 30g/L is hard without preheating. Do authors have any thoughts on that?
- Secondly, figure 6 is not clear either, some of the flask/bottles remained unlabeled, authors are advised to look carefully into it.
- Introduction Line 32-33 “Extraction and separation of resveratrol from cuspidatum is economical and widely used”. Given the cost of resveratrol commercially, I don’t think the extraction and separation are economical, if so, why didn’t the authors use conventional extraction of resveratrol rather than converting it from polydatin. Do authors have any idea or estimation of comparison between the cost of both procedures?
- Line 58-59. Several different beta-glucosidase enzymes have been isolated from bacteria, including some thermostable enzymes, I am therefore unable to conclude the statements regarding finding a suitable enzyme from bacteria.
- Results Lines, 75-76, Please indicate the methods by which molecular mass and PI were calculated.
- In characterization studies, do authors have used model substrate pNPG? If so, it should be clearly stated.
- Line 78, 79, and elsewhere, pNPG should be p
- Overall, in the manuscript, there are minor grammatical errors, which should be carefully addressed, for example, Line 196 “advantageous compared with other” should be advantageous as compared to others.
- Do the authors have any idea/ hypothesis about why the activity was slightly enhanced in the presence of Triton X-100 β-mercaptoethanol and imidazole.
- It would be better to include an SDS-PAGE purification figure for protein purification.
- Lines 60-64, please include references for the statement.
- Figure 5, effect temperature on the enzymatic activity of SpBGL1, the activity at 35 °C and 45 °C is higher than 40 °C, do the authors have any statement about the decrease of activity at 40 °C? I believe this is an error in data reporting. Authors are advised to revisit their data in this regard carefully.
Reviewer 2 Report
In this manuscript, a β-D-glucosidase gene from Sphingomonas sp. was cloned and expressed in Escherichia coli. Using the recombinant β- D-glucosidase for bioconversion of polydatin, the productivity of resveratrol was improved significantly. The manuscript can be considered for publication in Molecules after the following concerns are revised.
1. The abstract is not well written and should be rewritten. It does not adequately present the data from the result, and only the conversion rate and time are presented. When the conversion rate is mentioned, the concentration of the substrate should be indicated, otherwise, the higher conversion rate is meaningless. The same problem exists in the Discussion and Conclusion sections.
2. Page 2, line 35. "poludatin" should be "polydatin". There are some other errors in the text, please check the manuscript carefully.
3. In the introduction, it is not described why the β-D-glucosidase gene from Sphingomonas sp. was cloned for bioconversion of polydatin. I did see a previous study on the use of β-D-glucosidase from this fungus for the bioconversion of polygonin to resveratrol.
4. Page 2, line 81. Please check the unit of Vmax, usually it is μmol/(mg•min).
5. Page 4.The Y-axis of Figure 4 has no title.
6. Page 5, Figure 5. Why isn't the Y-axis presented by the conversion rate? It is better to state the substrate concentration in the text, and then the Y-axis of Figure 5 presented by the conversion rate directly, rather than the concentrations of the substrate and product, for the readers to calculate by themselves.
7. In section 2.4, the substrate concentration is 30 g/L with a purity of 20%, which is equal to a concentration of 6 g/L polydatin. 4.23 g/L of resveratrol was obtained after bioconversion, and the conversion rate calculated by the author is 95.3%. The MW of polydatin is 390.4, and MW of resveratrol is 228.24, so the conversion rate si (4.23/228.24)/(6/390.4)*100%=120.6%, is it correct?
8. Some references are not properly labeled, such as on the page 2 line 56, page 9 line 207, page 9 line 222. The references number should be placed after the author's name.
